# In Vitro Probitotic Evaluation of *Saccharomyces boulardii* with Antimicrobial Spectrum in a *Caenorhabditis elegans* Model

**DOI:** 10.3390/foods10061428

**Published:** 2021-06-20

**Authors:** Ramachandran Chelliah, Eun-Ji Kim, Eric Banan-Mwine Daliri, Usha Antony, Deog-Hwan Oh

**Affiliations:** 1Department of Food Science and Biotechnology, College of Agriculture and Life Science, Kangwon National University, Chuncheon 24341, Korea; ramachandran865@gmail.com (R.C.); rladmswldkssud@naver.com (E.-J.K.); ericdaliri@gmail.com (E.B.-M.D.); 2Department of Biotechnology and Food Technology, Anna University, Chennai 600 025, India; usha.antony@gmail.com

**Keywords:** yeast, probiotic, fermented food, antimicrobial, thermal process

## Abstract

In the present study, we screened for potential probiotic yeast that could survive under extreme frozen conditions. The antimicrobial and heat-stable properties of the isolated yeast strains *Saccharomyces boulardii* (*S. boulardii*) (*KT000032*, *KT000033*, *KT000034*, *KT000035*, *KT000036*, and *KT000037*) was analyzed and compared with commercial probiotic strains. The results revealed that the tested *S. boulardii KT000032* strain showed higher resistance to gastric enzymes (bile salts, pepsin, and pancreatic enzyme) at low pH, with broad antibiotic resistance. In addition, the strain also showed efficient auto-aggregation and co-aggregation abilities and efficient hydrophobicity in the in-vitro and in-vivo *C. elegens* gut model. Further, the KT000032 strain showed higher antimicrobial efficiency against 13 different enteropathogens and exhibited commensal relationships with five commercial probiotic strains. Besides, the bioactive compounds produced in the cell-free supernatant of probiotic yeast showed thermo-tolerance (95 °C for two hours). Furthermore, the thermo-stable property of the strains will facilitate their incorporation into ready-to-eat food products under extreme food processing conditions.

## 1. Introduction

Yeasts are eukaryotic microorganisms widespread in natural environments, including the normal microbial flora of humans, plants, airborne particles, water, food products, and in many other ecological niches. They are essential in many complex ecosystems, as frequent early colonizers of nutrient-rich substrates [1,2]. They are involved in broad interactions with other microorganisms, including symbiosis, mutualism, parasitism, and competition. It is a foreseeable part of the microflora of different fermented foods and beverages. Their habitat covers both human and animal origin, with a significant impact on food safety and organoleptic characteristics. Both baker’s and brewer’s yeasts (*Saccharomyces cerevisiae*) are available as dietary supplements because of their high nutrition and mineral content. Regardless of their non-human origin, such non-pathogenic yeasts fulfill the significant criteria for probiotic definition [3]. Probiotics currently in use are primarily common gram-positive lactic acid bacteria of the genera *Lactobacillus* and *Bifidobacterium*. Some yeast, such as *Saccharomyces boulardii* strains [4], are also used as nutritional supplements or pharmaceutical aid for therapeutic agents [5]. Some lactic acid bacterial (LAB) strains to play a significant role in the safety and quality of fermented products due to the production of secondary metabolites that act as antimicrobial agents and can prevent foodborne pathogens. Research studies confirmed that probiotic organisms such as *Bifidobacterium bifidum* and *Streptococcus thermophilus* reduced the incidence of acute diarrhea and rotavirus shedding [6,7]. The beneficial effect of other probiotic bacteria in the intestinal tract might be partly because of the induction of protective host antimicrobials. Certain antimicrobials, which are proteinaceous compounds synthesized during the metabolism of LAB., have antagonist activities [8]. Interest in probiotic yeast has risen not only in animal feed preparation but also for human applications. Yeasts are rarely associated with foodborne illness and based on their history, most of yeast species are recognized as safe by the European Food Safety Authority (EFSA) [9].

Studies have reported that *S**. boulardii* species had been proven to have antidote effects against various gastrointestinal diseases [10,11]. Hence, it is recognized as a non-bacterial prototype of a probiotic. Several mechanisms have been suggested for the broad health-promoting effects of consuming food grade yeasts [12,13]. Some of the reported effects of yeasts as probiotic organisms in clinical trials are (1) antibiotic-associated diarrhea; (2) infectious diarrhea including recurrent *Clostridium difficile*; (3) irritable bowel syndrome; and (4) inflammatory bowel diseases (IBD) [14]. *S. boulardii* efficacy was observed in ulcerative colitis and Crohn’s disease [15].

## 2. Materials and Methods

### 2.1. Phenotypic and Genotypic Identification

Idli batter (an Indian staple food) was prepared with the rice variety CR1009 and black gram dhal taken in the ratio 3:1 (*w*/*w*). They were washed with tap water individually and soaked in double the amount of drinking water at room temperature (27 ± 2 °C) for 4 h. Rice was coarsely ground and mixed with black gram dhal ground to a smooth batter with salt (2% *w*/*w*). The batter was mixed well with hand and allowed to ferment in an incubator at 30 C for 12 h. A total of s yeast strains of *S. boulardii* (KT000032, KT000033, KT000034, KT000035, KT000036, and KT000037), *Pichia kudriavzevii* (KT000037)] and 3 bacterial strains (*Escherichia coli*; *Enterococcus faecium* and *Lactobacillus casei*) were isolated from 12 h fermented idli batter based on colony morphology on yeast peptone dextrose (YPD) agar. The isolation was performed in the Food Microbiology Laboratory of Anna University. The yeast isolates were characterized using morphological and biochemical analysis [5]. The molecular identification of yeast was per-formed using specific 18S rRNA primers NS1 (5-GTAGTCATATGCTTGTCTC-3) and NS8 (5-TCCGCAGGTTCACCTACGGA-3). The sequencing was performed at Serene Biosciences (Bangalore), India. All the obtained sequences were searched using the Basic Local Alignment Search Tool (BLAST), and the sequences were registered in GenBank, NCBI. The current manuscript focuses only on the *S. boulardii* strains (KT000032, KT000033, KT000034, KT000035, KT000036, and KT000037). The probiotic characterization and heat stability with antimicrobial efficacy was quantified for the six strains. 

### 2.2. Commercial Probiotics

The isolated probiotic yeasts were tested against the commercially available probiotic strains such as *Lactobacillus reuteri* (Ecoflora), *Lactobacillus reuteri* (KT000042), *Saccharomyces boulardii* (Econorm-250 µg), *Lactobacillus rhamnosus* (GR7), and *Lactobacillus acidophilus* (MTCC111), using the well diffusion method to see if the isolated cultures are compatible against the commercial probiotics thus proving to be commensals. Likewise, the efficiency of commercial and isolated probiotics was tested against the pathogens.

### 2.3. Culture Technique

Isolated yeast and commercial probiotics were grown in an aerobic atmosphere for 48 h at 37 °C for bacterial strains and at 30 °C for yeast strains in static conditions. Pure colonies were picked and sub-cultured in 25 mL of yeast peptone dextrose (YPD) broth for 16 h. To eliminate traces of glucose from the culture medium, centrifugation at 8000× *g* (Eppendorf centrifuge 5804R, 2051 rotor, Seoul, Korea) for 20 min at 4 °C. The pellets were separated from the supernatant and suspended in sterile phosphate-buffered saline (PBS, 0.85%) of pH 7.4, followed by centrifugation. The shot was washed twice with sterile PBS. The optical densities were adjusted to 0.1 (595 nm), and these suspensions were immediately applied in analyzing the probiotic efficiency.

### 2.4. Acid and Alkaline Tolerance

The growth was evaluated by inoculating yeast strains (1% *v*/*v*) into YPD tubes adjusted to pH (1–7) (3.0 mol l-1 HCL, Merck Ltd., Seoul, Korea) and incubated at 30 °C for 48 h in static condition [16]. At periodical (0, 24, 48, 72, 96, 120 h) intervals, the cultures were serially diluted in PBS (pH 6) and spread-plated in YPD agar. The plates were incubated at 30 °C, and the colonies were counted using Lapiz colony counter. The viable population was determined in terms of the survival rate calculated based on Chen et al. [17]. % survival calculated based on the following formula = (Log number of viable cells survived (CFU mL-1)/Log number of initially viable cells inoculated) × 100.

### 2.5. Bile Tolerance

Yeast strains (1% *v*/*v*) were inoculated into fresh 25 mL YPD broth supplemented with different concentrations (0.1%, 0.3%, and 0.5% *w*/*v*) of bile salts (Sigma Aldrich, Mumbai, India), which were kept static at 30 °C. The bile tolerance of strains was evaluated based on estimating the number of viable cells after 0 and 72 h of incubation on YPD agar plates [17]. Bile tolerance % calculated based on the following formula = (Log number of viable cells in broth with bile/Log number of viable cells in broth without bile) × 100.

### 2.6. In Vitro Survival in Gastric Juice

To determine the gastric tolerance of yeast strains based on the method proposed by Psomas et al. [18] with slight modifications. Yeast strains (30 mL) were centrifuged at 8000× *g* for 20 min (4 °C). The cell pellet was washed with and resuspended in sterile PBS in the ratio of 1:9. Then, 0.1 mL of the suspension was added to 1.0 mL simulated gastric juice made up of a pepsin solution (0.3% *w*/*v*, P6887—Sigma Aldrich) and NaCl (0.5% *w*/*v*) to achieve pH 1.2. The yeast cells were enumerated on YPD agar plates after 0, 30, 60, 90, and 120 min incubation at 30 °C. The results were expressed as the decrease in viability [17]. Decrease in viability % calculated based on the following fomula = {[Log number of viable cells at 0 h (CFU mL^−1^)—Log number of viable cells after 2 h (CFU mL^−1^)]/Log number of viable cells at 0 h (CFU mL^−1^)} × 100.

### 2.7. Determination of Simulated Transit Tolerance

Pancreatin (Sigma Aldrich) was prepared at a concentration of (1 mg mL^−1^) and pepsin concentration of (3 mg mL^−1^). Yeast strains (1% (*v*/*v*)) were incubated at 30 °C for two hours under agitation, leading to simulated enteric phase 1. The pH was increased to 6.8–7.2, and the samples were incubated at 30 °C for two h under agitation, leading to simulated enteric phase 2 and reaching six h of the assay. The yeast strains were enumerated in aliquots collected from triplicate samples after two and four, and six hours. Aliquots of 1 mL were pour-plated into YPD agar [19].

### 2.8. Cell Surface Hydrophobicity

The separation of solvent and aqueous layers on the addition of hydrocarbons and subsequent adherence of the microbial cells in the aqueous layer instead of the solvent layer demonstrates that the strains can bind to intestinal epithelia in the human gut. The efficiency of the culture to adhere to hydrocarbons was tested using two different solvents, xylene, and toluene. The growth medium with the strains was centrifuged at 8000× *g* for 20 min at 4 °C, and the pellets were collected in 1 mL tubes. The pellets were washed twice with PUM buffer (K_2_HPO_4_, pH 6.5 ± 0.2). They were then resuspended and diluted in the same buffer to obtain an absorbance value ranging from 0.6 to 0.8 at 600 nm. Clear washed cell suspension (5 mL) was taken in a round-bottomed test tube along with 1 mL of xylene and toluene separately. The test tube containing the solution was agitated using a vortex mixer (REMI, Mumbai, India) for 2 min. Then it was incubated for 30 min and 1 h at 37 °C for phase separation. The transparent lower aqueous layer was separated from the upper solvent layer containing the cells. The absorbance of the two layers at 600 nm was noted, and the percentage hydrophobicity was calculated using the formula in [19]. Hydrophobicity % calculated based on the following formula = [1 − (Absorbance of solvent layer/Absorbance of the aqueous layer)] × 100

### 2.9. C. elegans Gut Colonization Assay

The yeast strains were screened for colonization of *C. elegans* gut (N2 stage, obtained from the Caenorhabditis Genetic Center (CGC), Minneapolis, MN, USA) based on the method of Lee et al. [20] with some minor methodology modifications. *C. elegans* were fed to individual yeast strains seeded on the Nematode Growing Medium (NGM) plates containing statin for 14 days. Every three days, the plates were re-seeded. The attachment of yeast was calculated based on the probability of total, five worms were randomly selected, further washed thrice with M9 buffer (to remove microbial strains attached outside the body of the worm), and placed on potato dextrose agar (PDA) (Sigma-Aldrich, Gangnam-gu, Seoul, Korea) plates containing 10% tatric acid. The worms were washed three times with M9 (M6030 buffer and homogenized using a mechanical homogenizer (BT704, BT Lab Systems Inc., Saint Louis, MO, USA) in a 1.5 mL Eppendorf tube containing M9 buffer supplemented with 1% Triton X-100, Sigma-Aldrich, Korea). The lysate was diluted serially (10-fold) in M9 buffer and plated on PDA agar (pH 5.5). The plates were incubated at 30 °C for 48 h and live

### 2.10. Antimicrobial Activity

The well-diffusion assay detected antagonist activity of lactobacilli and yeast strains based on Reinheimer et al. [21] using selected cultures of enteropathogens purchased from Ramachandra Medical College as follows: *Escherichia coli*, *Staphylococcus aureus*, *Enterococcus faecalis*, *Micrococcus luteus*, *Klebsiella pneumoniae*, *Salmonella typhi*, *Salmonella paratyphi A*, *Salmonella paratyphi B*, *Proteus mirabilis*, *Vibrio cholerae*, Shigella flexneri, *Shigella dysenteriae*, *Pesudomonas aeruginosa*. The overnight grown cultures were centrifuged for 20 min at 8000× *g* (4 °C). The supernatants were filtered through a 0.22 mm filter (Millipore, Gagny, France) to remove residual cells. Further, the supernatants (in 4 mm diameter wells) were tested for antagonist activity against the pathogens mentioned above plated onto Muller Hinton agar (Becton, Dickinson and Company Tullastrasse 8-12, Heidelberg, Germany). The antimicrobial activity was recorded as growth-free inhibition zones (diameter) around the well [22]. Briefly, each bacterial suspension (200 µL) was inoculated on Nutrient Agar (NA) by spread plat. Sterile paper discs of 6 mm in diameter each, loaded with different solvent ex-tracts of different concentrations (0.051, 0.0255, 0.01275, and 0.0051 mg/mL) were aseptically placed on the surface of the agar. To allow complete diffusion of CFS, the plates were allowed to stand for 10 min before incubation at 37 °C for 24 h. Control experiments were carried out under the same conditions by phosphate buffer saline (PBS) as a negative control. The growth kinetics studies were conducted by adding different concentrations of protein to pre-diluted overnight cultures (0.1 OD) to a final volume of 1 mL in a sterile cuvette, and incubating at 37 °C for 4–6 h, with continuous shaking. The OD at 600 nm was measured every 60 min by spectrophotometer (Biophotometer, Eppendorf Korea, Seoul, Korea), and values were recorded.

### 2.11. Aggregation and Co-Aggregation Assays

Yeast strains grown at 30 °C for 48 h in YPD broth were harvested by centrifugation at 8000× *g* for 20 min (4 °C), washed, and resuspended in sterile PBS to an optical density (OD) of 1% (1 × 10^7^ CFU mL^−1^) at 600 nm. For autoaggregation assays, suspensions of yeast strains (4 mL) was taken in glass test tubes and mixed by vortexing. Absorbance was measured immediately (A0), after five h and 24 h (At = 5 h, 24 h). Auto aggregation % calculated based on the following formula = [1 − (At/A0)] × 100. Auto-aggregation was monitored by phase-contrast microscopy at 100 times magnification after Gram staining [17].

For the co-aggregation assay, bacterial suspension was prepared in the same way as previously described. Equal volumes (2 mL) of probiotic strains and pathogen suspensions were divided into glass test tubes mixed using a cyclomixer (REMI). Control tubes contained 2 mL of a suspension of each strain. Absorbance was measured immediately, after five h and 24 h. Co-aggregation % calculated based on the following formula = {[(Ax + Ay)/2] − A(x + y)}/[(Ax + Ay)/2] × 100. (Where A represents absorbance, x and y represent each of the two strains in the control tubes and (x + y) their mixture) [17].

### 2.12. Susceptibility to Antibiotics

Isolated yeast strains were tested against 30 antibiotics with different modes of actions such as amikacin (MD001), amoxycillin (MD002), azithromycin (MD004), benzyl penicillin (MD062), cefalexin (cephalexin) (MD014), cefepime (MD070), cefotaxime (cephotaxime) (MD064), chloramphenicol (MD016), ciprofloxacin (MD017), erythromycin (MD022), gemifloxacin (MD076), gentamicin (MD061), kanamycin (MD026), levofloxacin (MD027), methicillin (MD031), moxifloxacin (MD033), neomycin (MD036), norfloxacin (MD038), ofloxacin (MD039), pefloxacin (MD040), polymyxin-B (MD043), rifampicin (MD045), roxithromycin (MD046), streptomycin (MD048), sulphadiazine (MD050), sulphamethizole (MD052), teicoplanin (MD055), vancomycin (MD060), tetracycline (MD056), meropenem (SD727) (Himedia, Mumbai, India) with standard antibiotic concentration previously determined by testing against pathogens [17]. The diameter of the inhibition zone was measured after 48 h of incubation at 30 °C.

### 2.13. Carbohydrates Fermentation Assay

The fermentation assay for fifteen dietary sugars and six sugar alcohols was tested. arabinose (Ar) (DD001), cellobiose (DD028) (Ce), dextrose (DD002) (De), fructose (DD017) (Fc), galactose (DD016) (Ga), inulin (DD026) (In), lactose (DD004) (La), maltose (DD005) (Ma), mannose (DD007) (Mo), melibiose (DD030) (Mb), raffinose (DD029) (Rf), rhamnose (DD010) (Rh), sucrose (DD013) (Su), trehalose (DD031) (Te), xylose (DD014) (Xy), adonitol (DD025) (Ad), dulcitol (DD003) (Du), inositol (DD027) (Is), mannitol (DD006) (Mn), salicin (DD011) (Sa), and sorbitol (DD012) (Sb). Andrade peptone water (5 mL) (without sugar supplemention) was mixed with a 25 mg disc (0.5%) of each carbohydrate in the corresponding sterile test tubes. The addition of yeast strains (2%) was followed by incubation at 37 °C and 30 °C in aerobic conditions and examined for color changes at 0, 12, 24, 36, 48, and 72 h. Triplicates of each combination (bacteria and carbohydrate) were performed. For the negative control, yeast was replaced by sterile saline [23].

### 2.14. Stability of Cell-Free Supernatant

To determine the thermostability, yeast strains (1 mL) were subjected to high temperatures at 95 ± 2 °C for 30, 60, 90, and 120 min and 121 °C for 15 min to determine their thermostability. The treated cells’ growth was observed by re-suspension in fresh 10 mL PDB and incubation at 30 °C for 48 h. The growth rate was calculated for 48 h based on the OD600, and the samples were collected at 12 h intervals. Likewise, the cell-free supernatant (CFS) was treated at high temperatures and tested for stability by measuring the antagonist activity against clinically isolated pathogenic *Escherichia coli* (a Gram-negative indicator strain) and *Staphylococcus aureus* (a Gram-positive bacterial strain) [24].

### 2.15. Statistical Analysis

All analytical tests were performed in triplicates. Mean ± standard deviation was calculated. Analysis of variance was done to obtain the differences’ significance to conclude meaningful generalizations [24].

## 3. Results and Discussion

The isolated yeast strains from the frozen idli batter were screened for probiotic characteristics based on exposure to gastrointestinal acidity and pepsin after oral consumption, and hence, the efficiency of potential probiotic candidates was evaluated. In addition, the antimicrobial efficacy and their heat-stable properties were screened for the novel probiotic enriched heat-stable product development.

### 3.1. Phenotypic and Genotypic Identification

The isolated yeast strains obtained from 12 h fermented idli batter were confirmed using methylene blue stain [25] and phenotypic profile (morphology, physiological and biochemical assay like metabolic activity and fermentation pattern) that remains the benchmark and an identity card for classification of a taxon group. Later, genotype and species level were determined using the universal primer (18S rRNA) for yeast, and it was identified as *S. boulardii* (KT000032, KT000033, KT000034, KT000035, KT000036, and KT000037) (Figure 1). The strains were deposited in GenBank, and the accession numbers obtained are given above. The isolated yeast strains showed a close relationship with each other.

### 3.2. Acid Tolerance

When isolated strains were subjected to pH changes (1–7), the yeast cultures were found to survive even at extreme pH ranges. At pH 1, all six isolated *S. boulardii* (KT000032, KT000033, KT000034, KT000035, KT000036, and KT000037) strains were unable to survive, but from pH 2, the yeast strains could survive and withstand till pH 7 (Table 1). Acid tolerance confirms that the isolated strains can survive in both stomach and intestine without getting degraded. Hence, they may be effectively used in tablets or also syrups. The pH values have been selected to incorporate the probiotic in food products that may be highly acidic or alkaline.

Further, the tolerance was assessed for 120 h to evaluate the probiotics’ stability in food products. Previous reports by Czerucka et al. [13] indicated that overexpression of genes related to protein synthesis and stress responses could contribute to the increased growth rate and better survival of *S. boulardii* in acidic pH. Similarly, yeast strains such as *Issatchenkia orientalis*, *Candida parapsilosis*, and *Candida albicans* were tested for tolerance at pH 1. 2 to 5 by suspending PBS buffer cultures’ survivability to pH 5 as reported by Psomas et al. [18].

### 3.3. Effects of Bile Salt on the Viability

All six isolated *S. boulardii* strains (KT000032, KT000033, KT000034, KT000035, KT000036, and KT000037) were found to survive in 0.1, 0.3, 0.5% bile salt for 72 h. Among the tested strains KT000032 showed a higher survival rate (99, 95 and 77%) (Table 2). On the other hand, Kourelis et al. [26] indicated that the strains *S**. cerevisiae* 982, *S. boulardii* KK1, and *Kluyveromyces lactis* (570, 630) exhibited a significantly higher capacity to survive at pH 3.0 in comparison with the other yeast strains. Hence, the yeast strains’ survival pattern exposed to various concentrations (0.1%, 0.15%, 0.3%, and 0.5% (*w*/*v*)) of bile salts showed variation among the ability of the test strains to tolerate or grow in the presence of bile salts.

### 3.4. Effects of Low pH and Gastric Juice on the Viability (Pancreatin Tolerance Test)

The efficiency of gastric tolerance were analyzed for the isolated *S. boulardii*, in that KT000032 strains showed a higher survival rate in pepsin with 1.2 pH after two h than in pancreatin bile salt solution at 8.0 pH after six h of incubation (Table 3). In addition, investigations of Ogunremi et al. [27] showed similar results for the strains *Pichia kluyveri* LKC17, *Issatchenkia orientalis* OSL11, *Pichia kudriavzevii* OG32, *Pichia kudriavzevii* ROM11, and *Candida tropicalis* BOM21 with a higher growth rate in simulated gastric juice (pH 2.0, pepsin) than in simulated intestinal conditions (pH 7.5, pancreatin).

### 3.5. Gut Colonization Assay

#### 3.5.1. Cell Surface Hydrophobicity—In Vitro

The hydrophobicity efficiency of *S. boulardii* (KT000032, KT000033, KT000034, KT000035, KT000036, and KT000037) in the presence of toluene and xylene ranging from 62 to 90% and 47 to 57%, respectively (Table 4) under identical conditions. Among the hydrocarbons, *S. boulardii* showed a strong affinity towards toluene due to its substantial electron donor property, which measures the ability to adhere to intestinal mucus as suggested by Wadstroum et al. [28].

#### 3.5.2. *C. elegans* Gut Colonization Ability—In Vivo Hydrophobicity Analysis

As shown in Figure 3, among the yeast strains *S. boulardii* (KT000032, KT000033, KT000034, KT000035, KT000036, and KT000037), specifically KT000032 showed high persistence in *C. elegans* gut than the other probiotic yeast candidates. *However*, *E. coli OP50 (control)* showed no attachment ability to *C. elegans* gut. Values from triplicate showed very mild variations (Figure 2). The attachment of probiotics with epithelial cells mucasal layer was a required trait, which aided colonization in the host gut and showed antimicrobial efficacy towards enteropathogens [29]. In this current study, the yeast KT000032 revealed substantial colonization efficiency at different feeding states with negligible differences. These results showed many similarities with the previous reports that showed that different lactic acid bacterial strains possess different attaching and colonization efficiency in *C. elegans* gut [30]. The correlation between colonization and hydrophobicity was reported by chelliah et al. [31], but some studies showed no correlation between the two functional properties [32]. However, in the current study, the probiotic yeast *S. boulardii* KT000032 showed efficient auto-aggregation compared with the other probiotic yeast strains. The outcome, of the results indicate the relationship between attachment ability and cell hydrophobicity. Thus, in vitro studies may not always imitate in vivo situations, accounting for our observation.

### 3.6. Antimicrobial Activity Determination

#### 3.6.1. Disc Diffusion Method

The cell-free supernatant (bioactive compounds) of *S. boulardii* (KT000032, KT000033, KT000034, KT000035, KT000036, and KT000037) was tested against the enteropathogenic bacterial group. The results revealed that after 24 and 48 h of incubation, the clearance zone showed similar antagonist activity against pathogens for commercial and isolated probiotic strains (Table 5). Among the 13 pathogens tested, *S. boulardii* (KT000032) showed highly susceptibility towards nine strains (*E. faecalis*, *M. luteus*, *K. pneumoniae*, *S. typhi*, *S. dysenteriae*, *V. cholerae*, *S. flexneri*, *P. mirabilis*, *S. paratyphi B*), but all the tested CFS showed less sensitivity towards *S. paratyphi A* and *P. aeruginosa*, respectively (Table 5). Czerucka et al. [13] reported a beneficial *S. boulardii* against various enteric pathogens such as *C. difficile*, *V. cholerae*, *Salmonella*, *Shigella*, and *E. coli*. Thus *S. boulardii* appeared to act by two main mechanisms: (i) production of factors that neutralized bacterial toxins and (ii) modulation of the host cell signaling pathway in pro-inflammatory response during bacterial infection.

#### 3.6.2. Growth Curve—Minimum Growth Inhibitory Concentration Determination

The antimicrobial activity of CFS of isolated yeast strains using the disc diffusion method, *S. boulardii* (KT000032) showed higher antibacterial activity. Further, the minimum inhibitory concentration was determined based on the growth curve method [31]. Further, the minimum inhibitory concentration (MIC) of different plant extracts was determined based on growth inhibitory assay [33,34] against indicator pathogens such as *E. coli*, *S. aureus*, *S. typi*, *S. dysenteriae, V. cholera* and *S. flexneri* (Figure 3).

### 3.7. Aggregation and Co-Aggregation

In-vitro evaluation of auto-aggregation and ability to co-aggregate with potential enteric pathogens can be used for preliminary screening and selection of the best probiotic strains among the S. boulardii (KT000032, KT000033, KT000034, KT000035, KT000036, and KT000037). The auto-aggregation rate of KT000032 strain showed higher efficacy after 24 h of incubation indicated 93.14% (Table 6). Likewise, the microscopic analysis further confirmed the clustering of cells and the presence of aggregates (Figure 4). In this study, among the tested S. boulardii KT000032 strains showed potential antimicrobial efficiency (Figure 3) towards prevention of intestinal colonization by pathogens based on in vitro co-aggregation with the tested pathogens (S. aureus, E. coli, and S. typhimurium) (Table 6). In addition, S. boulardii KT000032 showed a higher ability to co-aggregate with both gram-positive and negative bacteria, which may have potential applications. Higher co-aggregation efficiency was observed with S.typhimurium (65.07%) (Figure 4) followed by E. coli (47%) and S. aureus (41.16%, respectively). This observation is supported by a report that established certain pathogenic bacteria possess binding molecules on their surfaces that can bind to yeasts due to mannan and polysaccharides on their cell wall’s outer layer [35]. A previous report conducted by Ogunremi et al. [27] indicated that the test yeast strains’ autoaggregation ability increased throughout 24 h, especially Pichia kudriavzevii OG32 highest percentage auto-aggregation (91.85%) after 24 h. Sourabh et al. [36] also reported favorable aggregation abilities in yeast strains isolated from some Indian fermented food products. A previous study indicated that yeast strains showed co-aggregation abilities with the E. coli and S. flexneri. The highest co-aggregation ability (71.57%) was detected for P. kudriavzevii OG32 and E. coli. In comparison, the lowest (28.23%) was detected for P. kudriavzevii OG32 and S. flexneri after 24 h of co-incubation [27]. Microorganisms’ ability to adhere to epithelial cells and mucosal surfaces is critical for probiotic selection [37]. The adherent potentials of microbes correlate with the cell surface’s aggregation and hydrophobic properties [38].

### 3.8. Susceptibility of Antibiotics

All six strains of *S. boulardii* (KT000032, KT000033, KT000034, KT000035, KT000036, and KT000037) were tested against 30 different antibiotics with different modes of action such as inhibition of cell wall, nucleic acid, and protein synthesis. The results obtained confirmed that they showed resistance towards all the tested antibiotics (Table 7). Antibiotic resistance genes might be transferred between members of the resident gut flora and to and from transient bacterial probiotics. Czerucka et al. [13] reported that *S. boulardii* is naturally resistant to antibiotics and can be prescribed to patients under antibiotic treatment.

### 3.9. Sugar Fermentation Assay

The fermentation reactions occurred at a higher rate at 37 °C aerobic atmospheres than at 30 °C with some exceptions. The results from the fermentation assay are reported in Figure 5. *S. boulardii* (KT000032) strain showed less fermentation of rhamnose, raffinose, dulcitol, sorbitol, cellobiose, arabinose, inulin, sucrose, lactose, adonitol, maltose, salicin, mannitol, galactose, inositol and mannitol even after 48 h. This indicated weak reactions and pH values were between 5.2 and 6.8 after 72 h of incubation under shaking condition (Table 8), but xylose, fructose, mannose, dextrose, and trehalose were fermented in 48 h. Among the 21 sugars, xylose was utilized predominantly. The fermentation of sugars was similar at both 30 °C and 37 °C for *S. boulardii* (KT000032), but the growth rate had a negligible reduction initially for 12 h at 30 °C. Crittenden et al. [39] reported the probiotics *Lactobacillus brevis* and the intestinal bacteria *Bacteroides* spp. to ferment the dietary fibers β-glucan, xylan, xylooligosaccharides (XOS), and arabinoxylan.

### 3.10. Comparative Analysis of Antagonist Activity within Commercial and Isolated Probiotics

Commercial probiotics (L. reutri (Ecoflora), L. reuteri (KT000042), S. boulardii (Econorm), L. rhamnosus (GR7), L. acidophilus (MTCC 111)) were plated against CFS of six strains of S. boulardii (KT000032). The results indicated that no clearance zone was observed (Table 9) after 24 and 48 h of incubation. Thus, the isolated probiotic strains were compatible with commercial probiotics and can be used along with strains available in food and pharmaceuticals.

### 3.11. Thermo-Stability of the Yeasts

*S. boulardii* is a thermophile that has enzymes that can function at high temperatures. The *S. boulardii* strains (KT000032, KT000033, KT000034, KT000035, KT000036, and KT000037) were tested for survival at high temperatures, and it was observed that they were capable of survival at 95 °C for 2 h and also at 121 °C for 15 min (Figure 5). Microscopic endospore (budding within itself) formation was observed (Figure 5). The viability of the cells is shown in Table 10. The heat-treated six *S. boulardii* were re-inoculateion of 1% culture in YPD broth, and their growth rate was compared with the control. It was found that there was no significant change in the growth rate of heat-treated (95 °C) strain, but the autoclaved strain showed 50% decrease in the growth rate, which may be due to spore formation during the autoclaving temperature [40].

### 3.12. Stability of Cell-Free Supernatant (CFS)

The CFS was collected after 48 h of incubation of the isolated *S. boulardii* (KT000032) and commercially available probiotic strains in their respective medium and centrifuged at 3000× *g* for 10 min. It is enriched with bioactive compounds secreted by the probiotic bacteria as a by-product of the defense mechanism against the entero-pathogens. CFS from the probiotic yeast strains was stable after heat treatment at 95 °C for 15, 30, 60, 120 min and incubation at 30 °C and 15 °C for 24 h. Likewise, no loss of activity was observed after autoclaving the CFS at 121 °C for 15 min. The stability test was based on their activity against pathogens (*E. coli* and *S. aureus*, *S. typhi* and *S. dysenteriae*). The results revealed that the CFS of all the six strains did not lose their antimicrobial efficacy after heat treatment, and hence they are heat stable (Table 11). The zone formation observed after the neutralization by sodium bicarbonate indicated that the supernatant of the sample will not lose its antimicrobial activity in intestinal conditions and stomach conditions. A previous report by Nowroozi et al. [41] supports that the CFS of *L. plantarum*, *L. delbruekii*, *L. acidophilus*, *L. brevis* was stable after 5 days at 25 °C and showed antibacterial activity against S. aureus. The antibacterial activity was also stable at 100 °C for 10 min and 56 °C for 30 min. A small (<1 kDa) heat-stable and water-soluble anti-inflammatory molecule termed *Saccharomyces* anti-inflammatory factor (SAIF) identified in the yeast supernatant has been hypothesized to play a role in the antimicrobial effect of *S. boulardii* [13].

## 4. Conclusions

Research on probiotics has reported majorly on lactic acid bacteria (LAB) strains and fewer studies were conducted for probiotic yeast. In the current study among the isolated yeast strains, *S. boulardii* (KT000032) demonstrated promising probiotic characteristics with a higher survival rate under in vitro simulated gastric environmental conditions. Besides, *S. boulardii* (KT000032) strain produced secondary metabolites (bioactive compounds) that showed higher antimicrobial activity. In addition the tested cell-free supernatant and the strains showed thermostability. Hence, these effective probiotic yeast strains emerge as new tools for novel food manufacturers with higher viability under extreme storage conditions. Thus, the current research represents a breakthrough in incorporating heat stable probiotic yeast in ready-to-eat dairy, cereal, or pulse-based heatable food products. Further studies to ascertain the bioactive compounds of these probiotic yeast candidates are warranted.

## Figures and Tables

**Figure 1 foods-10-01428-f001:**
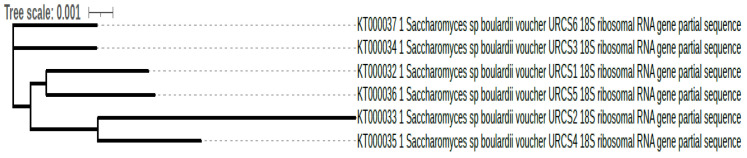
Phylogenetic tree based evolutionary relationships of the isolated probiotic yeasts from frozen idli batter.

**Figure 2 foods-10-01428-f002:**
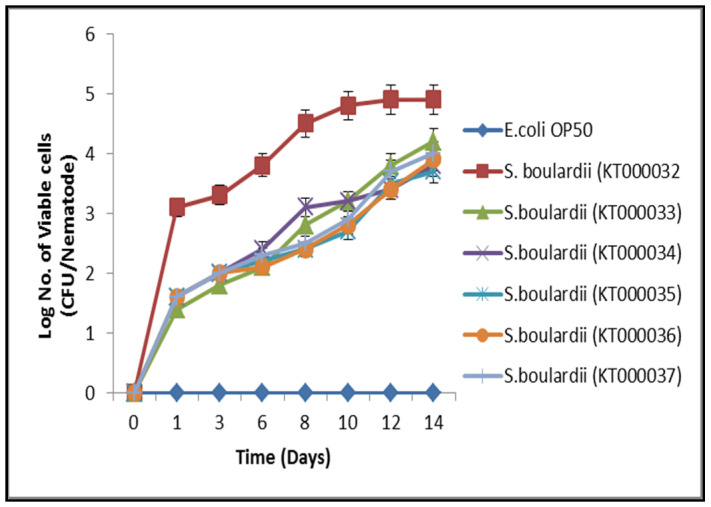
Bacterial colonization ability in *C. elegans*. The data points represent average CFU from five worms ± standard deviation of three independent experiments.

**Figure 3 foods-10-01428-f003:**
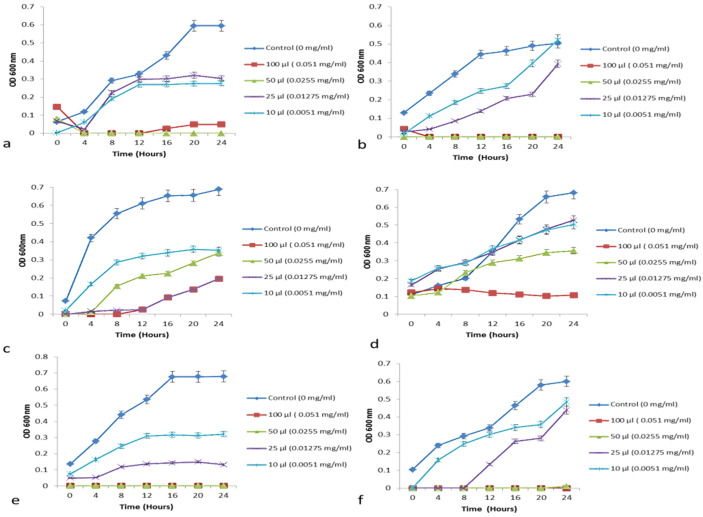
Antibacterial activities of *S. boulardii* (KT000032) cell free supernatant against the following pathogens (**a**) *E. coli* (**b**) *S. aureus* (**c**) *S. typi* (**d**) *S.dysenteriae* (**e**) *V. cholerae* (**f**) *S. flexneri*. Bars represent the means of three replicates (*n* = 3) ± SD. Data points with the same alphabets are not significantly different (*p* ˃ 0.05) using one-way ANOVA followed by Tukey’s test (ANOVA Software—New Version 2021.2).

**Figure 4 foods-10-01428-f004:**
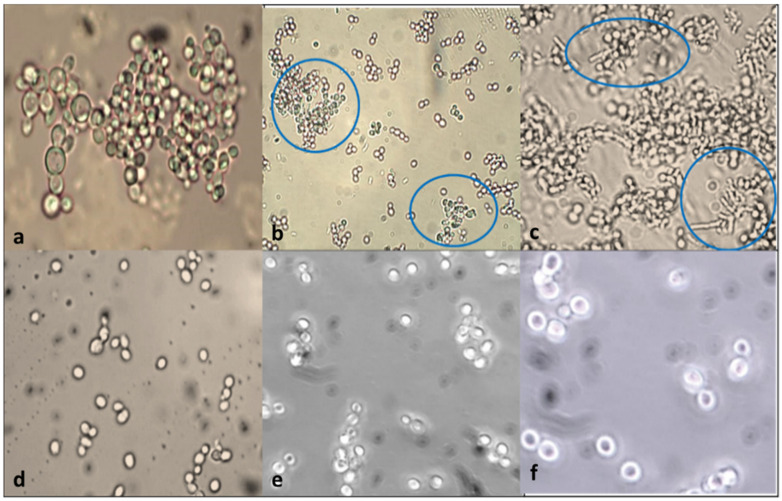
(**a**) Auto-aggregation of *S. boulardii* KT000032 grown in YPD broth. (**b**) Co-aggregation of *S. boulardii* and *S. aureus*. (**c**) Co-aggregation of *S. boulardii* and *E. coli* (phase-contrast microscope at 100 times magnification after methylene blue staining). (**d**) *S. boulardii* in PBS (control without heat treatment). (**e**) *S. boulardii* in PBS (heat-treated at 95 °C for 120 min). (**f**) *S. boulardii* in PBS (autoclaved at 121 °C for 15 min) endospore formation.

**Figure 5 foods-10-01428-f005:**
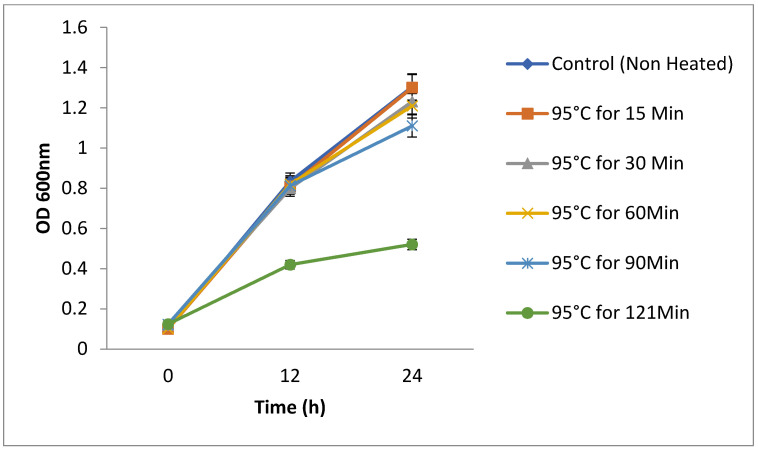
Heat Stability of yeast cells *S. boulardii* strains (KT000032) and post-growth efficiency with different time duration at 600 nm.

**Table 1 foods-10-01428-t001:** Acid tolerance of six *Saccharomyces boulardii* strains isolated from frozen idli batter in phosphate-saline buffer (pH 2.5).

Isolated Strains	pH	Survival (log CFU mL^−1^) at Different Time Periods (h)						Survival (%) at Different Time Periods (h)				
		0	24	48	72	96	120	24	48	72	96	120
*S. boulardii* (KT000032)	1	8.34 ± 0.10	<1.00	<1.00	<1.00	<1.00	<1.00	-	-	-	-	-
	2	8.36 ± 0.03	8.02 ± 0.02	7.33 ± 0.12	7.27 ± 0.09	7.09 ± 0.02	6.89 ± 0.12	95 b	87 c	86 c	84 d	82 d
	3	8.21 ± 0.32	8.16 ± 0.17	8.13 ± 0.12	7.75 ± 0.03	7.13 ± 0.19	7.12 ± 0.43	98 ^a^	98 ^a^	94 ^b^	87 ^c^	86 ^c^
	4	8.60 ± 0.08	8.52 ± 0.19	8.50 ± 0.32	8.21 ± 0.03	8.58 ± 0.31	8.00 ± 0.05	95 ^b^	99 ^a^	96 ^a^	89 ^c^	83 ^d^
	5	8.86 ± 0.11	8.42 ± 0.21	8.42 ± 0.05	8.29 ± 0.13	8.24 ± 0.04	6.20 ± 0.07	99 ^a^	99 ^a^	87 ^c^	87 ^c^	86 ^c^
	6	8.44 ± 0.22	8.23 ± 0.18	8.22 ± 0.18	8.11 ± 0.19	8.12 ± 0.01	8.04 ± 0.21	97 ^a^	100 ^a^	87 ^c^	87 ^c^	85 ^c^
	7	8.60 ± 0.08	8.52 ± 0.19	8.50 ± 0.32	8.21 ± 0.03	8.58 ± 0.31	8.00 ± 0.05	95 ^b^	99 ^a^	96 ^a^	89 ^c^	83 ^d^
*S. boulardii* (KT000033)	1	8.34 ± 0.10	<1.00	<1.00	<1.00	<1.00	<1.00	ND	ND	ND	ND	ND
	2	8.13 ± 0.03	4.06 ± 0.03	4.03 ± 0.02	4.45 ± 0.01	3.58 ± 0.31	2.68 ± 0.07	43 ^g^	38 ^h^	37 ^h^	34 ^i^	24 ^j^
	3	8.86 ± 0.11	8.42 ± 0.21	8.42 ± 0.05	8.29 ± 0.13	8.24 ± 0.04	6.20 ± 0.07	99 ^a^	99 ^a^	87 ^c^	87 ^c^	86 ^c^
	4	8.87 ± 0.16	8.39 ± 0.22	8.39 ± 0.21	8.11 ± 0.14	8.04 ± 0.31	6.86 ± 0.32	100 ^a^	100 ^a^	85 ^c^	85 ^c^	83 ^d^
	5	8.73 ± 0.02	8.57 ± 0.25	8.43 ± 0.06	8.45 ± 0.17	8.17 ± 0.36	7.12 ± 0.01	99 ^a^	98 ^a^	88 ^c^	85 ^c^	84 ^d^
	6	8.69 ± 0.02	8.41 ± 0.24	8.23 ± 0.15	8.76 ± 0.06	8.41 ± 0.16	6.21 ± 0.17	99 ^a^	97 ^a^	92 ^b^	88 ^c^	76 ^e^
	7	8.83 ± 0.02	8.26 ± 0.26	8.19 ± 0.02	8.38 ± 0.10	8.23 ± 0.08	7.15 ± 0.12	97	97 ^a^	88 ^c^	87 ^c^	86 ^c^
*S. boulardii* (KT000034)	1	8.34 ± 0.10	<1.00	<1.00	<1.00	<1.00	<1.00	ND	ND	ND	ND	ND
	2	8.16 ± 0.14	8.18 ± 0.01	8.15 ± 0.02	8.23 ± 0.43	8.12 ± 0.04	7.23 ± 0.05	97 ^a^	98 ^a^	87 ^c^	86 ^c^	77 ^e^
	3	8.44 ± 0.22	8.23 ± 0.18	8.22 ± 0.18	8.11 ± 0.19	8.12 ± 0.01	8.04 ± 0.21	97 ^a^	100 ^a^	87 ^c^	87 ^c^	85 ^c^
	4	8.92 ± 0.04	8.33 ± 0.20	8.33 ± 0.01	8.12 ± 0.12	8.05 ± 0.32	7.79 ± 0.41	99 ^a^	99 ^a^	86 ^c^	85 ^c^	83 ^d^
	5	8.44 ± 0.14	8.27 ± 0.27	8.19 ± 0.02	7.63 ± 0.19	7.12 ± 0.31	7.04 ± 0.10	99 ^a^	98 ^a^	92 ^b^	87 ^c^	75 ^e^
	6	8.12 ± 0.17	8.07 ± 0.28	7.32 ± 0.20	7.32 ± 0.21	7.19 ± 0.40	6.20 ± 0.04	97 ^a^	89	89 ^c^	88 ^c^	77 ^e^
	7	8.73 ± 0.02	8.57 ± 0.25	8.43 ± 0.06	8.45 ± 0.17	8.17 ± 0.36	7.12 ± 0.01	99 ^a^	98 ^a^	88 ^c^	85 ^c^	84 ^d^
*S. boulardii* (KT000035)	1	8.34 ± 0.10	<1.00	<1.00	<1.00	<1.00	<1.00	ND	ND	ND	ND	ND
	2	8.78 ± 0.22	8.05 ± 0.23	8.05 ± 0.03	8.33 ± 0.11	8.32 ± 0.05	6.32 ± 0.06	98 ^a^	98	90 ^b^	90 ^b^	79 ^e^
	3	8.69 ± 0.02	8.41 ± 0.24	8.23 ± 0.15	8.76 ± 0.06	8.41 ± 0.16	6.21 ± 0.17	99 ^a^	97 ^a^	92 ^b^	88 ^c^	76 ^e^
	4	8.73 ± 0.02	8.57 ± 0.25	8.43 ± 0.06	8.45 ± 0.17	8.17 ± 0.36	7.12 ± 0.01	99 ^a^	98 ^a^	88 ^c^	85 ^c^	84 ^d^
	5	8.83 ± 0.02	8.26 ± 0.26	8.19 ± 0.02	8.38 ± 0.10	8.23 ± 0.08	7.15 ± 0.12	97 ^a^	97 ^a^	88 ^c^	87 ^c^	86 ^c^
	6	8.44 ± 0.14	8.27 ± 0.27	8.19 ± 0.02	7.63 ± 0.19	7.12 ± 0.31	7.04 ± 0.10	99 ^a^	98 ^a^	92 ^b^	87 ^c^	75 ^e^
	7	8.12 ± 0.17	8.07 ± 0.28	7.32 ± 0.20	7.32 ± 0.21	7.19 ± 0.40	6.20 ± 0.04	97 ^a^	89 ^c^	89 ^c^	88 ^c^	77 ^e^
*S. boulardii* (KT000036)	1	8.34 ± 0.10	<1.00	<1.00	<1.00	<1.00	<1.00	ND	ND	ND	ND	ND
	2	8.26 ± 0.13	8.48 ± 0.29	7.32 ± 0.21	7.12 ± 0.22	7.31 ± 0.23	6.11 ± 0.13	99 ^a^	87 ^c^	85 ^c^	76 ^e^	74 ^f^
	3	8.73 ± 0.02	8.57 ± 0.25	8.43 ± 0.06	8.45 ± 0.17	8.17 ± 0.36	7.12 ± 0.01	99 ^a^	98 ^a^	88 ^c^	85 ^c^	84 ^d^
	4	8.83 ± 0.02	8.26 ± 0.26	8.19 ± 0.02	8.38 ± 0.10	8.23 ± 0.08	7.15 ± 0.12	97 ^a^	97 ^a^	88 ^c^	87 ^c^	86 ^c^
	5	8.44 ± 0.14	8.27 ± 0.27	8.19 ± 0.02	7.63 ± 0.19	7.12 ± 0.31	7.04 ± 0.10	99 ^a^	98 ^a^	92 ^b^	87 ^c^	75 ^e^
	6	8.12 ± 0.17	8.07 ± 0.28	7.32 ± 0.20	7.32 ± 0.21	7.19 ± 0.40	6.20 ± 0.04	97 ^a^	89 ^c^	89 ^c^	88 ^c^	77 ^e^
	7	8.26 ± 0.13	8.48 ± 0.29	7.32 ± 0.21	7.12 ± 0.22	7.31 ± 0.23	6.11 ± 0.13	99 ^a^	87 ^c^	85 ^c^	76 ^e^	74 ^e^
*S. boulardii* (KT000037)	1	8.34 ± 0.10	<1.00	<1.00	<1.00	<1.00	<1.00	ND	ND	ND	ND	ND
	2	8.69 ± 0.02	8.41 ± 0.24	8.23 ± 0.15	8.76 ± 0.06	8.41 ± 0.16	6.21 ± 0.17	99 ^a^	97 ^a^	92 ^b^	88 ^c^	76 ^e^
	3	8.73 ± 0.02	8.57 ± 0.25	8.43 ± 0.06	8.45 ± 0.17	8.17 ± 0.36	7.12 ± 0.01	99 ^a^	98 ^a^	88 ^c^	85 ^c^	84 ^d^
	4	8.83 ± 0.02	8.26 ± 0.26	8.19 ± 0.02	8.38 ± 0.10	8.23 ± 0.08	7.15 ± 0.12	97 ^a^	97 ^a^	88 ^c^	87 ^c^	86 ^c^
	5	8.69 ± 0.02	8.41 ± 0.24	8.23 ± 0.15	8.76 ± 0.06	8.41 ± 0.16	6.21 ± 0.17	99 ^a^	97 ^a^	92 ^b^	88 ^c^	76 ^e^
	6	8.73 ± 0.02	8.57 ± 0.25	8.43 ± 0.06	8.45 ± 0.17	8.17 ± 0.36	7.12 ± 0.01	99 ^a^	98 ^a^	88 ^c^	85 ^c^	84 ^d^
	7	8.60 ± 0.08	8.52 ± 0.19	8.50 ± 0.32	8.21 ± 0.03	8.58 ± 0.31	8.00 ± 0.05	95 ^b^	99 ^a^	96 ^a^	89 ^c^	83 ^d^

Values are expressed in mean ± standard deviation (*n* = 3) Different superscripts (a, b, c, d, e, f, g, h, i, j) represent significantly different values (*p* < 0.05), (non-detected:ND).

**Table 2 foods-10-01428-t002:** Bile tolerance of six *Saccharomyces boulardii* strains isolated from frozen idli batter.

Isolated Strains	Survival (log CFU mL^−1^) at 0 h	Survival at Different Bile Salt (oxgall) Concentrations (% *w*/*v*)					
		0.1 (log CFU mL^−1^)	%	0.3 (log CFU mL^−1^)	%	0.5 (log CFU mL^−1^)	%
*S. boulardii* (KT000032)	8.12 ± 0.07	8.09 ± 0.07	99 ^a^	7.75 ± 0.16	95 ^d^	6.22 ± 0.09	77 ^i^
*S. boulardii* (KT000033)	8.36 ± 0.10	8.12 ± 0.17	97 ^c^	7.37 ± 0.31	88 ^g^	5.78 ± 0.21	69 ^k^
*S. boulardii* (KT000034)	8.62 ± 0.03	8.36 ± 0.24	97 ^c^	7.56 ± 0.12	88 ^g^	5.66 ± 0.18	66 ^m^
*S. boulardii* (KT000035)	8.35 ± 0.22	8.16 ± 0.18	98 ^b^	7.42 ± 0.24	89 ^f^	5.76 ± 0.41	69 ^k^
*S. boulardii* (KT000036)	8.49 ± 0.07	8.31 ± 0.07	98 ^b^	7.83 ± 0.13	92 ^e^	6.11 ± 0.04	72 ^j^
*S. boulardii* (KT000037)	8.72 ± 0.07	8.44 ± 0.23	97 ^c^	7.45 ± 0.18	85 ^h^	5.89 ± 0.12	67 ^l^

Tolerance of lactic acid bacteria to bile salt (0.3%). Values are expressed as the mean ± standard deviation (*n* = 3). Different superscripts (a, b, c, d, e, f, g, h, i, j, k, l, m) represent significantly different values (*p* < 0.05).

**Table 3 foods-10-01428-t003:** Gastric juice and intestinal transit tolerance of six *Saccharomyces boulardii* strains isolated from frozen *idli* batter.

Isolated Strains	Survival (log CFU mL^−1^) at 0 h	Survival after 2 h in Pepsin Solution (pH 1.2) (3 mg mL^−1^)		Survival after 6 h in Pancreatin Solution (pH 8.0) (1 mg mL^−1^)	
		log CFU mL^−1^	%	log CFU mL^−1^	%
*S. boulardii* (KT000032)	8.72 ± 0.07	7.41 ± 0.16	85 ^b^	7.63 ± 0.11	94 ^a^
*S. boulardii* (KT000033)	8.36 ± 0.10	7.28 ± 0.26	87 ^b^	5.38 ± 0.31	64 ^e^
*S. boulardii* (KT000034)	8.62 ± 0.03	7.31 ± 0.45	85 ^b^	5.67 ± 0.52	66 ^b^
*S. boulardii* (KT000035)	8.35 ± 0.22	7.44 ± 0.18	89 ^b^	5.58 ± 0.23	67 ^d^
*S. boulardii* (KT000036)	8.49 ± 0.07	7.17 ± 0.13	84 ^c^	5.48 ± 0.17	65 ^d^
*S. boulardii* (KT000037)	8.12 ± 0.07	5.46 ± 0.22	63 ^e^	5.11 ± 0.27	63 ^b^

Viability of lactic acid bacteria in the presence of simulated intestinal fluid (pepsin (pH = 1.2) and 1 mg/mL pancreatin in 0.85% NaCl, *w*/*v*, pH 8.0). Values are expressed as the mean ± standard deviation (*n* = 3). Different superscripts (a, b, c, d, e) represent significantly different values (*p* < 0.05).

**Table 4 foods-10-01428-t004:** Surface hydrophobicity of *Saccharomyces boulardii* strains with two different solvents (xylene, toluene).

Isolated Strains	Xylene Layer (A_600nm_)	Aqueous Layer (A_600nm_)	%	Toluene Layer (A_600nm_)	Aqueous Layer (A_600nm_)	%
*S. boulardii* (KT000032)	0.85± 0.0.3	0.67 ± 0.24	79 ^c^	0.81± 0.33	0.73 ± 0.22	90 ^a^
*S. boulardii* (KT000033)	0.87± 0.07	0.41 ± 0.18	47 ^f^	0.65 ± 0.23	0.37 ± 0.08	57 ^e^
*S. boulardii* (KT000034)	0.69 ± 0.06	0.33 ± 0.14	48 ^f^	0.77± 0.13	0.55 ± 0.21	71 ^c^
*S. boulardii* (KT000035)	0.58 ± 0.09	0.27 ± 0.32	47 ^f^	0.73± 0.12	0.45 ± 0.13	62 ^d^
*S. boulardii* (KT000036)	0.83 ± 0.13	0.47 ± 0.16	57 ^e^	0.84 ± 0.06	0.66 ± 0.12	90 ^a^
*S. boulardii* (KT000037)	0.78 ± 0.14	0.38 ± 0.13	49 ^g^	0.81± 0.16	0.72 ± 0.17	89 ^b^

Values are expressed in mean ± standard deviation (*n* = 3) Different superscripts (a, b, c, d, e, f, g) represent significantly different values (*p* < 0.05).

**Table 5 foods-10-01428-t005:** Antimicrobial activity (mm) of *S. boulardii* strains CFS against commercial probiotics and clinical pathogens.

Dysentery-Causing Clinical Pathogens	Isolated Strains					
	Commercial Probiotics					
	*S. boulardii* (KT000032)	*S. boulardii* (KT000033)	*S. boulardii* (KT000034)	*S. boulardii* (KT000035)	*S. boulardii* (KT000036)	*S. boulardii* (KT000037)
	*L. reuteri* (Ecoflora)	*L. reuteri* (KT000042)	*S. boulardii* (Econorm 250 µg)	*L. rhamnosus* (GR7)	*L. acidophilus* (MTCC111)	
*E. coli*	22 ± 0.04 ^b^	15 ± 0.08 ^c^	15 ± 0.01 ^c^	16 ± 0.11 ^c^	14 ± 0.07 ^d^	26 ± 0.12 ^a^
	22 ± 0.13 ^b^	21 ± 0.04 ^b^	20 ± 0.05 ^b^	22 ± 0.13 ^b^	26 ± 0.04 ^a^	
*S. aureus*	25 ± 0.4 ^a^	15 ± 0.05 ^c^	18 ± 0.22 ^c^	11 ± 0.16 ^b^	18 ± 0.11 ^c^	19 ± 0.45 ^c^
	24 ± 0.2 ^b^	19 ± 0.14 ^c^	25 ± 0.19 ^a^	17 ± 0.07 ^c^	20 ± 0.17 ^b^	
*E. faecalis*	21 ± 0.31 ^b^	18 ± 0.07 ^c^	22 ± 0.65 ^b^	15 ± 0.16 ^a^	12 ± 0.52 ^b^	17 ± 0.67 ^c^
	17 ± 0.11 ^c^	17 ± 0.11 ^c^	17 ± 0.21 ^c^	14 ± 0.15 ^d^	18 ± 0.69 ^c^	
*M. luteus*	20 ± 0.14	15 ± 0.16 ^a^	16 ± 0.47 ^c^	17 ± 0.25 ^c^	11 ± 0.51 ^b^	14 ± 0.89 ^d^
	14 ± 0.6 ^d^	19 ± 0.18 ^c^	18 ± 0.07 ^c^	16 ± 0.21 ^c^	20 ± 0.27 ^b^	
*K. pneumoniae*	14 ± 0.01 ^c^	17 ± 0.01 ^c^	11 ± 0.66 ^b^	15 ± 0.56 ^c^	22 ± 0.07 ^b^	16 ± 0.58 ^c^
	19 ± 0.22 ^c^	22 ± 0.03 ^b^	15 ± 0.81 ^c^	19 ± 0.78 ^c^	11 ± 0.16 ^d^	
*S. typhi*	20 ± 0.01 ^b^	15 ± 0.11 ^b^	15 ± 0.01 ^a^	17 ± 0.31 ^c^	24 ± 0.07 ^b^	22 ± 0.23 ^b^
	25 ± 0.17 ^a^	21 ± 0.13 ^b^	13 ± 0.1 ^d^	22 ± 0.25 ^b^	16 ± 0.12 ^c^	
*S. paratyphi A*	13 ± 0.07 ^d^	17 ± 0.14 ^a^	12 ± 0.31 ^b^	15 ± 0.67 ^c^	17 ± 0.21 ^c^	25 ± 0.31 ^a^
	20 ± 0.15 ^a^	22 ± 0.03 ^b^	19 ± 0.25	25 ± 0.07 ^a^	21 ± 0.82 ^b^	
*S. paratyphi B*	21 ± 0.44 ^c^	20 ± 0.05 ^b^	15 ± 0.27 ^a^	18 ± 0.80 ^c^	21 ± 0.51 ^b^	16 ± 0.38 ^c^
	20 ± 0.07 ^b^	21 ± 0.21 ^b^	26 ± 0.21 ^a^	21 ± 0.37 ^b^	15 ± 0.68 ^a^	
*P. mirabilis*	21 ± 0.08 ^b^	16 ± 0.25 ^c^	21 ± 0.18 ^b^	17 ± 0.82 ^c^	15 ± 0.23 ^a^	19 ± 0.51 ^c^
	16 ± 0.04 ^c^	22 ± 0.27 ^b^	22 ± 0.19	19 ± 0.55 ^c^	17 ± 0.36 ^c^	
*V. cholerae*	25 ± 0.08 ^a^	14 ± 0.13 ^d^	10 ± 0.16 ^b^	15 ± 0.29 ^c^	17 ± 0.07	22 ± 0.22 ^b^
	14 ± 0.03 ^d^	25 ± 0.51 ^a^	25 ± 0.68 ^a^	13 ± 0.27 ^b^	15 ± 0.41 ^c^	
*S. flexneri*	22 ± 0.02 ^d^	18 ± 0.41 ^d^	18 ± 0.35 ^c^	17 ± 0.11 ^c^	15 ± 0.16 ^c^	25 ± 0.69 ^a^
	18 ± 0.02 ^d^	26 ± 0.26 ^a^	21 ± 0.21 ^b^	17 ± 0.18 ^c^	18 ± 0.13 ^c^	
*S. dysenteriae*	23 ± 0.44 ^b^	12 ± 0.07 ^b^	10 ± 0.37 ^b^	14 ± 0.36 ^d^	21 ± 0.13 ^b^	21 ± 0.51 ^b^
	22 ± 0.07 ^b^	19 ± 0.48 ^c^	17 ± 0.27 ^c^	18 ± 0.42 ^c^	27 ± 0.38 ^a^	
*P. aeruginosa*	22 ± 0.9 ^b^	11 ± 0.62 ^b^	11 ± 0.19 ^d^	16 ± 0.27 ^c^	14 ± 0.48 ^d^	21 ± 0.78 ^b^
	21 ± 0.015 ^b^	27 ± 0.04 ^a^	19 ± 0.16 ^c^	15 ± 0.23 ^c^	22 ± 0.41 ^b^	

Values are expressed as the mean ± standard deviation (*n* = 3). Different superscripts (a, b, c, d) represent significantly different values (*p* < 0.05).

**Table 6 foods-10-01428-t006:** Auto-aggregation and co-aggregation abilities of potential probiotic *Saccharomyces boulardii* strains after 20 h incubation at 37 °C.

Isolated Strains	Autoaggregation (%)		Coaggregation (%) with *E. coli*		Coaggregation (%) with *S. aureus*		Coaggregation (%) with *S. typhimurium*	
	5 h	24 h	5 h	24 h	5 h	24 h	5 h	24 h
*S. boulardii* (KT000032)	78.13 ± 0.11 ^g^	93.14 ± 0.31 ^b^	35.04 ± 0.12	47.14 ± 0.17 ^j^	30.14 ± 0.06 ^j^	41.16 ± 0.06 ^h^	45.3 ± 0.23 ^m^	65.07 ± 0.16 ^k^
*S. boulardii* (KT000033)	47.18 ± 0.12 ^f^	78.11 ± 0.21 ^c^	18.17 ± 0.04 ^l^	24.10 ± 0.14 ^k^	21.17 ± 0.03 ^k^	37.41 ± 0.07 ^h^	15.37 ± 0.31 ^l^	21.08 ± 0.12 ^k^
*S. boulardii* (KT000034)	42.14 ± 0.22 ^g^	84.33 ± 0.23 ^b^	20.25 ± 0.11 ^k^	21.07 ± 0.15 ^k^	27.11 ± 0.07 ^j^	32.15 ± 0.14 ^i^	16.28 ± 0.18 ^l^	24.31 ± 0.03 ^k^
*S. boulardii* (KT000035)	49.12 ± 0.07 ^f^	85.18 ± 0.05 ^a^	16.26 ± 0.15 ^l^	26.13 ± 0.11 ^j^	25.09 ± 0.02 ^j^	35.12 ± 0.08 ^h^	15.41 ± 0.15 ^l^	20.17 ± 0.15 ^k^
*S. boulardii* (KT000036)	68.12 ± 0.18 ^e^	79.26 ± 0.07 ^c^	17.46 ± 0.04 ^l^	28.08 ± 0.11 ^j^	24.15 ± 0.01 ^l^	38.33 ± 0.08 ^h^	16.42 ± 0.32 ^l^	21.07 ± 0.13 ^k^
*S. boulardii* (KT000037)	45.12 ± 0.17 ^d^	85.21 ± 0.28 ^a^	18.31 ± 0.09	22.16 ± 0.26 ^k^	28.13 ± 0.06 ^j^	36.33 ± 0.14 ^h^	17.43 ± 0.26 ^l^	24.23 ± 0.12 ^k^

Values are expressed in mean ± standard deviation (*n* = 3). Different superscripts (a, b, c, d, e, f, g, h, i, j, k, l, m) represent significantly different values (*p* < 0.05).

**Table 7 foods-10-01428-t007:** Antibiotic susceptibility (mm) of *Saccharomyces boulardii* strains CFS against commercial probiotics and clinical pathogens.

Antibiotic Susceptibility Based on Disc Diffusion Method (mm)							
Antibiotics	Isolated Strains						
	Commercial Probiotics						
	Dysentery Causing Clinical Pathogens						
	*L. reuteri* (Ecoflora)	*L. reuteri* (KT000042)	*S. boulardii* (Econorm 250 µg)	*L. rhamnosus* (GR7)	*L. acidophilus* (MTCC111)	*S. boulardii* (KT000032)	
	*E. coli*	*S. aureus*	*E. faecalis*	*M. luteus*	*K. pneumoniae*	*S. typhi*	*S. paratyphi A*
	*S. paratyphi B*	*P. mirabilis*	*V. cholera*	*S. flexneri*	*S. dysentriae*	*P. aeruginosa*	
Amikacin	R	R	R	R	R	R	
	17 ± 0.18	14 ± 0.25	16 ± 0.42	19 ± 0.03	22 ± 0.02	25 ± 0.08	21 ± 0.01
	19 ± 0.35	17 ± 0.12	15 ± 0.16	18 ± 0.08	25 ± 0.12	17 ± 0.42	
Amoxycillin	R	R	R	R	R	R	
	19 ± 0.02	26 ± 0.02	22 ± 0.01	25 ± 0.03	21 ± 0.05	17 ± 0.05	22 ± 0.12
	21 ± 0.27	22 ± 0.25	15 ± 0.05	23 ± 0.18	21 ± 0.16	19 ± 0.06	
Azithromycin	R	R	R	R	R	R	
	13 ± 0.03	19 ± 0.01	26 ± 0.08	22 ± 0.01	25 ± 0.05	21 ± 0.03	20 ± 0.08
	18 ± 0.02	20 ± 0.25	11 ± 0.01	16 ± 0.25	21 ± 0.42	25 ± 0.25	
Benzyl penicillin	R	R	R	R	R	R	
	R	9 ± 0.01	R	9 ± 0.03	R	12 ± 0.01	R
	07 ± 0.08	14 ± 0.42	R	8 ± 0.16	9 ± 0.12	9 ± 0.03	
Cefalexin	R	R	R	R	R	R	
	25 ± 0.02	17 ± 0.01	15 ± 0.03	18 ± 0.08	20 ± 0.08	11 ± 0.03	16 ± 0.12
	21 ± 0.16	19 ± 0.02	21 ± 0.12	22 ± 0.42	25 ± 0.01	26 ± 0.25	
Cefepime	R	R	R	R	R	R	
	21 ± 0.02	22 ± 0.22	21 ± 0.22	22 ± 0.02	25 ± 0.05	26 ± 0.06	19 ± 0.16
	18 ± 0.08	20 ± 0.12	11 ± 0.03	16 ± 0.05	21 ± 0.16	25 ± 0.05	
Cefotaxime	R	R	R	R	R	R	
	14 ± 0.05	16 ± 0.03	19 ± 0.05	22 ± 0.02	25 ± 0.08	21 ± 0.05	20 ± 0.08
	11 ± 0.42	16 ± 0.08	21 ± 0.42	25 ± 0.42	17 ± 0.05	15 ± 0.12	
Chloramphenicol	R	R	R	R	R	R	
	18 ± 0.01	20 ± 0.02	11 ± 0.03	16 ± 0.05	21 ± 0.22	25 ± 0.02	17 ± 0.03
	15 ± 0.25	18 ± 0.42	22 ± 0.16	25 ± 0.06	26 ± 0.01	19 ± 0.42	
Ciprofloxacin	R						
	R	R	R	R	R	R	
	25 ± 0.03	17 ± 0.02	15 ± 0.05	18 ± 0.06	20 ± 0.02	11 ± 0.05	16 ± 0.42
	21 ± 0.42	19 ± 0.02	22 ± 0.25	25 ± 0.03	21 ± 0.08	19 ± 0.16	
Erythromycin	R	R	R	R	R	R	
	14 ± 0.02	16 ± 0.22	19 ± 0.03	22 ± 0.06	25 ± 0.04	21 ± 0.42	18 ± 0.02
	20 ± 0.42	11 ± 0.02	16 ± 0.42	21 ± 0.23	25 ± 0.01	17 ± 0.18	
Gemifloxacin	R	R	R	R	R	R	
	20 ± 0.16	16 ± 0.04	14 ± 0.01	18 ± 0.04	22 ± 0.69	21 ± 0.05	22 ± 0.06
	24 ± 0.14	17 ± 0.03	14 ± 0.18	19 ± 0.16	25 ± 0.08	27 ± 0.02	
Gentamicin	R	R	R	R	R	R	
	15 ± 0.05	23 ± 0.42	21 ± 0.05	19 ± 0.04	17 ± 0.02	9 ± 0.06	22 ± 0.11
	19 ± 0.02	17 ± 0.08	19 ± 0.08	22 ± 0.02	21 ± 0.06	22 ± 0.07	
Kanamycin	R	R	R	R	R	R	
	18 ± 0.05	20 ± 0.21	11 ± 0.21	16 ± 0.01	21 ± 0.65	25 ± 0.16	17 ± 0.05
	15 ± 0.04	18 ± 0.16	20 ± 0.06	11 ± 0.02	16 ± 0.03	21 ± 0.38	
Levofloxacin	R	R	R	R	R	R	
	18 ± 0.05	20 ± 0.16	11 ± 0.02	16 ± 0.06	21 ± 0.38	25 ± 0.14	17 ± 0.05
	15 ± 0.02	18 ± 0.05	21 ± 0.25	25 ± 0.21	17 ± 0.02	15 ± 0.16	
Methicillin	R	R	R	R	R	R	
	21 ± 0.01	19 ± 0.06	17 ± 0.42	9 ± 0.42	22 ± 0.01	19 ± 0.01	17 ± 0.01
	17 ± 0.38	14 ± 0.16	16 ± 0.22	19 ± 0.21	22 ± 0.23	25 ± 0.21	
Moxifloxacin	R	R	R	R	R	R	
	18 ± 0.42	20 ± 0.06	11 ± 0.05	16 ± 0.11	21 ± 0.05	25 ± 0.21	17 ± 0.21
	15 ± 0.38	18 ± 0.05	25 ± 0.03	17 ± 0.02	15 ± 0.16	18 ± 0.08	
Neomycin	R	R	R	R	R	R	
	22 ± 0.42	17 ± 0.05	14 ± 0.05	16 ± 0.03	19 ± 0.02	22 ± 0.01	25 ± 0.02
	18 ± 0.02	20 ± 0.01	11 ± 0.38	16 ± 0.03	21 ± 0.04	25 ± 0.21	
Norfloxacin	R	R	R	R	R	R	
	18 ± 0.05	20 ± 0.38	11 ± 0.05	16 ± 0.22	21 ± 0.12	25 ± 0.04	17 ± 0.03
	15 ± 0.06	17 ± 0.16	15 ± 0.03	18 ± 0.01	17 ± 0.21	15 ± 0.04	
Ofloxacin	R	R	R	R	R	R	
	18 ± 0.01	20 ± 0.05	11 ± 0.42	16 ± 0.05	21 ± 0.05	25 ± 0.38	17 ± 0.05
	16 ± 0.42	19 ± 0.38	22 ± 0.04	25 ± 0.02	21 ± 0.16	19 ± 0.42	
Pefloxacin	R	R	R	R	R	R	
	17 ± 0.03	14 ± 0.02	16 ± 0.42	19 ± 0.05	22 ± 0.06	25 ± 0.04	21 ± 0.03
	19 ± 0.04	29 ± 0.06	22 ± 0.42	25 ± 0.07	26 ± 0.16	19 ± 0.01	
Polymyxin-B	R	R	R	R	R	R	
	14 ± 0.06	16 ± 0.14	19 ± 0.03	29 ± 0.06	25 ± 0.01	21 ± 0.22	19 ± 0.06
	16 ± 0.04	21 ± 0.25	25 ± 0.01	17 ± 0.04	15 ± 0.25	15 ± 0.01	
Rifampicin	R	R	R	R	R	R	
	21 ± 0.8	22 ± 0.01	15 ± 0.42	23 ± 0.04	21 ± 0.14	17 ± 0.06	14 ± 0.22
	16 ± 0.25	19 ± 0.14	22 ± 0.18	2519 ± 0.06	21 ± 0.38	19 ± 0.25	
Roxithromycin	R	R	R	R	R	R	
	17 ± 0.14	14 ± 0.02	16 ± 0.22	19 ± 0.03	22 ± 0.04	25 ± 0.01	21 ± 0.03
	19 ± 0.02	R	21 ± 0.42	22 ± 0.25	19 ± 0.06	23 ± 0.04	
Streptomycin	R	R	R	R	R	R	
	21 ± 0.16	22 ± 0.16	21 ± 0.22	22 ± 0.04	25 ± 0.01	26 ± 0.16	19 ± 0.14
	22 ± 0.12	17 ± 0.04	14 ± 0.01	16 ± 0.38	19 ± 0.04	22 ± 0.15	
Sulphadiazine	R	R	R	R	R	R	
	15 ± 0.03	18 ± 0.22	20 ± 0.02	19 ± 0.06	16 ± 0.11	21 ± 0.02	19 ± 0.08
	22 ± 0.02	29 ± 0.06	17 ± 0.38	15 ± 0.25	18 ± 0.03	23 ± 0.17	
Sulphamethizole	R	R	R	R	R	R	
	25 ± 0.14	21 ± 0.22	18 ± 0.04	20 ± 0.26	11 ± 0.04	16 ± 0.38	29 ± 0.06
	25 ± 0.25	17 ± 0.42	14 ± 0.38	16 ± 0.43	19 ± 0.25	22 ± 0.14	
Teicoplanin	R	R	R	R	R	R	
	20 ± 0.22	11 ± 0.04	16 ± 0.02	21 ± 0.12	17 ± 0.22	17 ± 0.04	19 ± 0.01
	22 ± 0.03	19 ± 0.02	17 ± 0.16	19 ± 0.06	22 ± 0.16	21 ± 0.02	
Vancomycin	R	R	R	R	R	R	
	R	R	R	R	7 ± 0.01	19 ± 0.06	8 ± 0.01
	R	R	9 ± 0.16	R	R	R	
Tetracycline	R	R	-	R	R		
	25 ± 0.14	17 ± 0.22	19 ± 0.06	18 ± 0.02	15 ± 0.03	18 ± 0.38	20 ± 0.01
	19 ± 0.06	16 ± 0.03	21 ± 0.01	19 ± 0.04	22 ± 0.01	19 ± 0.06	
Meropenem	R	R	R	R	R	R	
	17 ± 0.01	9 ± 0.38	29 ± 0.06	19 ± 0.1	17 ± 0.16	19 ± 0.1	22 ± 0.22
	21 ± 0.16	22 ± 0.42	22 ± 0.01	25 ± 0.11	21 ± 0.14	18 ± 0.01	

Viability of *S. boulardii* strains and commercial Lactic Acid Bacteria (LAB) to antibiotics (R: resistant). Values are expressed as the mean ± standard deviation (*n* = 3).

**Table 8 foods-10-01428-t008:** The carbohydrate fermentation reactions of *Saccharomyces boulardii* strains after an incubation of 48 h at 37 °C, of various bacteria. Differentiation discs were tested using Phenol Red Broth Base (M054). (−) negative reaction (pH > 6.8); (±) weak reaction (pH 5.2 to 6.8); (+) positive reaction (pH < 5.2) after 24, 48, and 72 h, respectively.

S. No.	Sugar Fermentation	*S. boulardii* (KT000032)	
		Fermentation of Sugars at 37 °C	Fermentation of Sugars at 30 °C
1	Raffinose (Rf)	−−−	−−−
2	Rhamnose (Rh)	−−−	−−−
3	Dulcitol (Du)	−−−	−−−
4	Sorbitol (Sb)	−−−	−−−
5	Cellobiose (Ce)	−−−	−−−
6	Xylose (Xy)	−−−	−−−
7	Arabinose (Ar)	−−−	−−−
8	Inulin (In)	−−−	−−−
9	Fructose (Fc)	−−±	−−−
10	Mannose (Mo)	−−±	−−−
11	Sucrose (Su)	−−−	−−−
12	Lactose (La)	−−±	−−−
13	Adonitol (Ad)	−−±	−−−
14	Dextrose (De)	−−±	−−±
15	Maltose (Ma)	−−±	−−±
16	Salicin (Sa)	−−−	−−−
17	Mannitol (Mn)	−−−	−−−
18	Trehalose (Te)	±±±	±±±
19	Galactose (Ga)	−−±	−−±
20	Inositol (Is)	−−−	−−−
21	Melibiose (Mb)	−−−	−−−

**Table 9 foods-10-01428-t009:** Commensal efficiency (disc diffusion method, mm) of CFS of *S. boulardii* strains against commercial probiotics and clinical pathogens.

Isolated Strains	Commercial Probiotics (mm)				
	*L. reuteri* (Ecoflora)	*L. reuteri* (KT000042)	*S. boulardii* (Econorm 250 µg)	*L. rhamnosus* (GR7)	*L. acidophilus* (MTCC111)
*S. boulardii* (KT000032)	ND	ND	ND	ND	ND

Viability of commercial probiotic against CFS of probiotic yeast, (non-detected: ND).

**Table 10 foods-10-01428-t010:** Heat stability of *S. boulardii* strains towards heat treatment at 95 °C and 121 °C with different periods.

Isolated Strains	Survival (log CFU mL^−1^) at 0 h	Survival (log CFU mL^−1^) at 95 °C (mm)				Survival (log CFU mL^−1^) at 121 °C
		15 min	30 min	60 min	120 min	15 min
*S. boulardii* (KT000032)	8.72 ± 0.07 ^a^	8.51 ± 0.12 ^a^	7.16 ± 0.16 ^b^	6.89 ± 0.06 ^c^	6.23 ± 0.06 ^c^	5.86 ± 0.03 ^d^

Stability of *S. boulardii* strains heated at 90 °C for 15, 30, 60, 120 min and 121 °C for 30 min. Values are expressed as the mean ± standard deviation (*n* = 3). Different superscripts (a, b, c, d) represent significantly different values (*p* < 0.05).

**Table 11 foods-10-01428-t011:** Thermostability of *S. boulardii* strain cell free supernatant (CFS) heated at 90 °C for 15, 30, 60, 120 min and 121 °C for 30 min. Values are expressed as the mean ± standard deviation (*n* = 3). Different superscripts (a, b, c, d) represent significantly.

CFS of the Strains	Zone of Inhibition (mm)						Clinical Pathogens
	95 °C				30 °C	15 °C	
	15 min	30 min	60 min	120 min	24 h	24 h	
*S. boulardii* (KT000032)	23 ^b^	23 ^b^	21 ^b^	19 ^c^	23 ^b^	23 ^b^	*E. coli*
	20 ^c^	20 ^c^	19 ^c^	19 ^c^	21 ^c^	21 ^c^	*S. aureus*
	21 ^b^	21 ^b^	19 ^c^	18 ^c^	22 ^c^	21 ^c^	*S. typhi*
	18 ^c^	18 ^c^	18 ^c^	15 ^c^	18 ^c^	18 ^c^	*S. dysenteriae*

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
