# Peer review of "In Vitro Probiotic Evaluation of Saccharomyces boulardii with Antimicrobial Spectrum in a Caenorhabditis elegans Model"

_foods, 2021, doi:10.3390/foods10061428_

Round 1

Reviewer 1 Report

The paper “Validation of potential probiotic and heat-stable properties of Saccharomyces boulardii from frozen idli batter: In vitro and Caenorhabditis elegans model analysis” is interesting for the topic focused on the identification of yeast probiotic strains. The work needs a thorough review. English is very poor, making it difficult to understand the sentence at times, as well as numerous grammatical errors.

Going into the merits, the main comments concern:

-The title is too long and I suggest making it easier

-Introduction: Reduce the part that talks about probiotic bacteria as the work is focused on yeasts. At the end of the introduction, add a sentence that explains the purpose of the work

Methods:

-The section 2.11 should be moved after the 2.1

- The experimental techniques are well explained, but the cellular concentration (CFU) used is not always clear or reported. Saying 1 gram is not enough: the authors explain in detail the CFU for each experiment.

Results:

- Line 358 Growth curve. The explanation of the method used should be moved to the methods section. Here, being in the results, just say the first two lines (359,360) and the final ones (368-370.)

- The tables must be redone and presented differently in order to make the results clear. For example, as regards the table 2d, it is better and more understandable to place the sugars vertically and horizontally the strains behaviour at the two different temperatures.

- Figure 5 should be deleted

- List tables in sequence 1, 2, 3 etc, not 1a, 1b, 1c ...

- The legends of the tables and figures must be detailed, for example the explanation of the abbreviations. Thus, in the legend of table 2d the abbreviations of sugars must be reported with the corresponding sugar.

Minor comments:

Line 129 K2HPO4   Correct in K2HPO4

Line 170 1x109 CFu ml-1 Correct in   1x109 CFu ml-1

Line 328 Put in italics S. boulardii 

Line 413 Put in italics S. aureus

Line 434 It seems strange that S. boulardii ferments sucrose poorly. Are there any references in the literature reporting a low ability of S. boulardii to ferment sucrose?

Author Response

Respond to reviewer comments 1

  1. The paper “Validation of potential probiotic and heat-stable properties of Saccharomyces boulardiifrom frozen idli batter: In vitro and Caenorhabditis elegans model analysis” is interesting for the topic focused on the identification of yeast probiotic strains. The work needs a thorough review. English is very poor, making it difficult to understand the sentence at times, as well as numerous grammatical errors.

The authors are really grateful for the reviewer valuable suggestion, towards formatting the manuscript in a readable format for the readers.

Going into the merits, the main comments concern:

  1. -The title is too long and I suggest making it easier

As per the reviewer valuable suggestion the title has been edited and cut shorted

  1. -Introduction: Reduce the part that talks about probiotic bacteria as the work is focused on yeasts. At the end of the introduction, add a sentence that explains the purpose of the work

As per the reviewers valuable suggestion the sentence related to probiotic bacteria were reduced in the manuscript

  1. Methods:
  2. -The section 2.11 should be moved after the 2.1
    • The experimental techniques are well explained, but the cellular concentration (CFU) used is not always clear or reported. Saying 1 gram is not enough: the authors explain in detail the CFU for each experiment.

As per the reviewers valuable suggestion the section 2.11 should be moved after the 2.1 in the manuscript materials and method section

In addition as per the reviewer suggestion the CFU ( cell concentration were explained in detail)

  1. Results:
    • Line 358 Growth curve. The explanation of the method used should be moved to the methods section. Here, being in the results, just say the first two lines (359,360) and the final ones (368-370.)

As per the reviewer valuable suggestion the material section of antimicrobial activity were edited and separated

(Section – 2.10 and 3.6.2)

  • The tables must be redone and presented differently in order to make the results clear. For example, as regards the table 2d, it is better and more understandable to place the sugars vertically and horizontally the strains behavior at the two different temperatures.

As per the reviewer valuable suggestion the table has been edited and incorporated in the manuscript.

  • Figure 5 should be deleted

As per the reviewer valuable suggestion the Figure 5 was deleted

  • List tables in sequence 1, 2, 3 etc, not 1a, 1b, 1c ...

As per the valuable suggestion of the reviewer the tables has been re ordered sequentially in the results and discussion section

  • The legends of the tables and figures must be detailed, for example the explanation of the abbreviations. Thus, in the legend of table 2d the abbreviations of sugars must be reported with the corresponding sugar.

As per the reviewers suggestion the legends has been edited in detail and the abbreviations were explained

Table 8. The carbohydrate fermentation reactions of Saccharomyces boulardii strains after an incubation of 48 hours at 37°C, of various bacteria. Differentiation discs were tested using Phenol Red Broth Base (M054).

*Symbols in the table indicate: -, negative reaction (pH > 6.8); ±, weak reaction (pH 5.2 to 6.8); +, positive reaction (pH < 5.2) after 24, 48, and 72 h, respectively.

  1. Minor comments:
  2. Line 129 K2HPO4   Correct in K2HPO4

The subscript has been edited in the formula

  1. Line 170 1x109CFu ml-1 Correct in   1x109 CFu ml-1

The superscript has been edited in the formula

  1. Line 328 Put in italics  boulardii

The scientific names has been provided with italics

  1. Line 413 Put in italics  aureus

The scientific names has been provided with italics

  1. Line 434 It seems strange that  boulardiiferments sucrose poorly. Are there any references in the literature reporting a low ability of S. boulardii to ferment sucrose?

Mainly the stain(S.boulardii(KT000032)), which we applied in the study, not able to utilize the sucrose , but most of the published manuscripts indicates sucrose is the main source of the growth of yeast  cell by producing invertase enzyme and hydrolyze them into glucose and fructose. But our strain (S.boulardii (KT000032)) possibly failed to produce invertase enzyme to hydrolyze, hence that may be one of the reson the sucrose was not utilized.

(Extracellular hydrolysis of sucrose allows other cells to share glucose and fructose.

Sucrose is hydrolyzed into glucose and fructose by invertase located in the cell wall. The glucose and fructose are imported into the cell by hexose transporters or escape into the medium by diffusion. (B) The glucose and fructose monosaccharides diffuse away from the cell wall and are more easily shared between cells)

Reviewer 2 Report

Comments on the submitted article:
Validation of potential probiotic and heat‐stable properties of Saccharomyces boulardii from frozen
idli batter: In vitro and Caenorhabditis elegans model analysis
Ramachandran Chelliah, Eun Ji Kim, Eric Banan‐Mwine Daliri, Usha Antony, and Deog Hwan Oh
General comments:
The possibility of using probiotic yeasts for the formulation of novel foods is studied in this article. Yeasts are isolated from idli batter and then a series of tests are performed in order to highlight the properties of interest of the isolated yeasts. Even if this article may be of interest for the development of such ingredients, it is at this stage very inadequate and major revisions are needed. Basically, the method of isolation of the six yeasts selected is not explained or commented on: why such a matrix was chosen; what are their origins (commercial; artisanal; same matrix or several?). Moreover, strains of Saccharomyces boulardii are isolated and then studied, but in Figure 1, it is not clear whether the three bacteria and the yeast (Pichia) positioned on the phylogenetic tree are also isolated or just used for positioning. Subsequently, all the tests performed often lack references and in particular the other strains used (LAB or pathogenic bacteria) are not described or justified. Not all tests are clearly described and conditions justified (e.g. different centrifugation methods to recover cells; why? etc.). Results are commented on too briefly and sometimes inappropriately (comparisons of results when
results are not yet presented). Throughout the manuscript there are errors (missing words) and the English needs to be revised. Figures are overall of poor quality; and some tables could be put in supplementary data (table 2c for example).
Specific remarks:
Introduction:
L 57 "this yeast" ‐ clarify as S boulardii is mentioned well above in the text.
L60‐64 ‐ the sentence must be rephrased
Materials and Methods :
L67 ‐ origin of idli batter to be specified
L78 ‐ which “bacterial strains”? put the paragraph presenting them before (2‐11)
L105 ‐ why different centrifugation methods?
L123‐124 ‐ reference of this technique?
L127 ‐ centrifugation conditions still different. Why?
L155 ‐ Lactobacilli ‐ introduce them at the beginning (and not after ‐> 2‐11)
L175 ‐ missing reference
L 182 ‐ to be put above (presentation of strains used) ‐ include all strains used.
L‐ 202 ‐ in the results, “growth rate” values are presented in figure5 but how are they obtained?
L 226 ‐ stat analysis ‐ ANOVA with which software?
Results and discussion
L232 ‐ "based on the are exposer" ‐ a word is missing?
L 234‐235 ‐ specify "heatstable product"?
Figure 1 ‐ position of the 3 bacteria and Pichia? Figure of very poor quality
L 255 ‐ the strain KT000033 is more sensitive to pH2 than the others (table 1‐a) ‐> review comment
L 260‐263 ‐ comment not relevant here.
L 276‐278 – reference to an article where pH effect is studied when there is a bile effect which should
be comment here ‐ review
L 314 ‐ anaysis ‐> analysis
L317 ‐ put a reference in M&M to control (E. coli OP50)
L 328 ‐ comments on aggregation when results are not yet presented.
L329‐330 ‐ a word is missing? Rephrase the sentence
L 358 – There is no comment on this paragraph (3.6.2)?
L427 ‐ table 2c ‐> suggestion to suppl. data
L430 ‐ paragraph 3‐9 to be revised. Comments on the ability to ferment sugars at different
temperatures are not clear. Lack of explanation on how to calculate growth rates (in fact final Optical
density? So it is not a growth rate).
L 464 ‐ review effect of heat treatment: effect on residual population? No effect on growth is
observed but are the inoculation rates (with viable cells) re‐ajusted? To be clarified.
L 474 ‐ comment not relevant
L 482 ‐ figure 6 ‐ line correspondences are not clear ‐ condition where speed + week is which
condition? (95°C ‐ 90 min? but symbols do not match).
L 493 ‐ "heating activity" meaning?

Author Response

Respond to reviewer comments 2

  1. The possibility of using probiotic yeasts for the formulation of novel foods is studied in this article. Yeasts are isolated from idli batter and then a series of tests are performed in order to highlight the properties of interest of the isolated yeasts. Even if this article may be of interest for the development of such ingredients, it is at this stage very inadequate and major revisions are needed.
  • The authors are really grateful for the reviewer valuable suggestion, towards formatting the manuscript in a readable format for the readers.
  1. Basically, the method of isolation of the six yeasts selected is not explained or commented on: why such a matrix was chosen; what are their origins (commercial; artisanal; same matrix or several?). Moreover, strains of Saccharomyces boulardii are isolated and then studied, but in Figure 1, it is not clear whether the three bacteria and the yeast (Pichia) positioned on the phylogenetic tree are also isolated or just used for positioning.
  • A total of 7 yeast strains of boulardii (KT000032, KT000033, KT000034, KT000035, KT000036, and KT000037), Pichia kudriavzevii (KT000037)] and 3 bacterial strains (Escherichia coli; Enterococcus faecium and Lactobacillus casei) were isolated from 12 h fermented idli batter based on colony morphology on yeast peptone dextrose (YPD) agar. The isolation was performed in the Food Microbiology Laboratory of Anna University. The yeast isolates were characterized using morphological and biochemical analysis based on van der Aa Kühle et al. (2001). The molecular identification of yeast was per-formed using specific 18S rRNA primers NS1 (5-GTAGTCATATGCTTGTCTC-3) and NS8 (5-TCCGCAGGTTCACCTACGGA-3). The sequencing was performed at Serene Biosci-ences, Bangalore, India. All the obtained sequences were searched using the Basic Local Alignment Search Tool (BLAST), and the sequences were registered in GenBank, NCBI.
  • The current manuscript focus only on the boulardii (KT000032, KT000033, KT000034, KT000035, KT000036, and KT000037) among the 6 strains the probiotic characterization , heat stability with antimicrobial efficacy was quantified
  • Hence when the phylogenetic was drawn we incorporated the bacterial and yeast strains together to show the relationship with each other.
  1. Subsequently, all the tests performed often lack references and in particular the other strains used (LAB or pathogenic bacteria) are not described or justified. Not all tests are clearly described and conditions justified (e.g. different centrifugation methods to recover cells; why? etc.).

As per the valuable reviewer suggestion the reference and the methodology section was edited uniformly for clear understanding for the readers

  • Acid and alkaline tolerance [16]; Bile Tolerance[17]; In vitro survival in gastric juice[17]; Determination of simulated transit tolerance[19]; Cell surface hydrophobicity[19]; elegans gut colonization assay[20]; Antimicrobial activity[21;22]; Aggregation and co-aggregation assays [17]; Susceptibility to antibiotics [17]; Carbohydrates fermentation assay [23]; Stability of cell-free supernatant [24]; Statistical analysis [24]
  • Centrifugation methods to recover cells – 8000g for 20 min (4°C)

  1. Results are commented on too briefly and sometimes inappropriately (comparisons of results when results are not yet presented). Throughout the manuscript there are errors (missing words) and the English needs to be revised. Figures are overall of poor quality; and some tables could be put in supplementary data (table 2c for example).

The authors are really grateful for the reviewer valuable suggestion, towards formatting the manuscript in a readable format for the readers. The tables has been re-ordered sequentially in the results and discussion section

Specific remarks:
Introduction:
L 57 "this yeast" ‐ clarify as S boulardii is mentioned well above in the text.

As per the valuable reviewer suggestion throughout the manuscript, scientific names has been provided with italics
L60‐64 ‐ the sentence must be rephrased

As per the valuable reviewer suggestion the sentence were rephrased
Materials and Methods :
L67 ‐ origin of idli batter to be specified

As per the reviewer valuable suggestion the origin of idli batter mentioned in detail

Idli batter (Indian staple food) was prepared with the rice variety CR1009 and black gram dhal taken in the ratio 3:1 (w/w). They were washed with tap water individually and soaked in double the amount of drinking water at room temperature (27 ± 2 C) for 4 h. Rice was coarsely ground and mixed with black gram dhal ground to a smooth batter with salt (2% w/w). The batter was mixed well with hand and allowed to ferment in an incubator at 30 C for 12 h.
L78 ‐ which “bacterial strains”? put the paragraph presenting them before (2‐11)

As per the reviewer valuable suggestion the details of isolated bacterial and yeast strain has been provided in the  2.1. Phenotypic and genotypic identification

  • A total of 7 yeast strains of S. boulardii (KT000032, KT000033, KT000034, KT000035, KT000036, and KT000037), Pichia kudriavzevii (KT000037)] and 3 bacterial strains (Escherichia coli; Enterococcus faecium and Lactobacillus casei) were isolated from 12 h fermented idli batter based on colony morphology on yeast peptone dextrose (YPD) agar. The isolation was performed in the Food Microbiology Laboratory of Anna University. The yeast isolates were characterized using morphological and biochemical analysis based on van der Aa Kühle et al. (2001). The molecular identification of yeast was per-formed using specific 18S rRNA primers NS1 (5-GTAGTCATATGCTTGTCTC-3) and NS8 (5-TCCGCAGGTTCACCTACGGA-3). The sequencing was performed at Serene Biosci-ences, Bangalore, India. All the obtained sequences were searched using the Basic Local Alignment Search Tool (BLAST), and the sequences were registered in GenBank, NCBI.
  • The current manuscript focus only on the S. boulardii (KT000032, KT000033, KT000034, KT000035, KT000036, and KT000037) among the 6 strains the probiotic characterization , heat stability with antimicrobial efficacy was quantified
  • Hence when the phylogenetic was drawn we incorporated the bacterial and yeast strains together to show the relationship with each other.
    L105 ‐ why different centrifugation methods?

As per the valuable reviewer suggestion the reference and the methodology section was edited uniformly for clear understanding for the readers

  • Centrifugation methods to recover cells – 8000g for 20 min (4°C)
    L123‐124 ‐ reference of this technique?

As per the valuable reviewer suggestion the reference and the methodology section was edited uniformly for clear understanding for the readers

  • Cell surface hydrophobicity[19]

L127 ‐ centrifugation conditions still different. Why?

As per the valuable reviewer suggestion the reference and the methodology section was edited uniformly for clear understanding for the readers

  • Centrifugation methods to recover cells – 8000g for 20 min (4°C)
    L155 ‐ Lactobacilli ‐ introduce them at the beginning (and not after ‐> 2‐11)

As per the valuable reviewer suggestion the corrections has been effect in the manuscript
L175 ‐ missing reference

As per the valuable reviewer suggestion the reference has been edited in the manuscript

Aggregation and co-aggregation assays [17]
L 182 ‐ to be put above (presentation of strains used) ‐ include all strains used.

As per the reviewers valuable suggestion the section 2.11 should be moved after the 2.1 in the manuscript materials and method section

L‐ 202 ‐ in the results, “growth rate” values are presented in figure5 but how are they obtained?

As per the reviewers valuable suggestion the Growth Kinetics were incorporated in 2.10 section – Antimicrobial activity

The growth kinetics studies were conducted by adding different concentrations of protein to pre-diluted overnight cultures (0.1 OD) to a final volume of 1 mL in a sterile cuvette, and incubating at 37 ◦C for 4–6 h, with continuous shaking. The OD at 600 nm was measured every 60 min by spectrophotometer (Biophotometer, Eppendorf Korea., Gangnam-gu Seoul, Korea), and values were recorded.
L 226 ‐ stat analysis ‐ ANOVA with which software?

As per the reviewers valuable suggestion the ANOVA software version incorporated

ANOVA Software - New Version 2021.2
Results and discussion
L232 ‐ "based on the are exposer" ‐ a word is missing?

As per the reviewers valuable suggestion the corrections has been effected in the manuscript

The isolated yeast strains from the frozen idli batter were screened for Probiotic characteristics based on the exposer towards gastrointestinal acidity and pepsin after oral consumption, and hence, the efficiency of potential probiotic candidates was evaluated. In addition, the antimicrobial efficacy and their heat-stable properties were screened for the novel probiotic enriched heat-stable product development
L 234‐235 ‐ specify "heatstable product"?

As per the reviewers valuable suggestion the corrections has been effected in the manuscript

In addition, the antimicrobial efficacy and their heat-stable properties were screened for the novel probiotic enriched heat-stable product development
Figure 1 ‐ position of the 3 bacteria and Pichia? Figure of very poor quality

As per the reviewers valuable suggestion the corrections has been effected in the manuscript

Figure 1 was edited and incorporated

L 255 ‐ the strain KT000033 is more sensitive to pH2 than the others (table 1‐a) ‐> review comment

S.boulardii (KT000032) was found to be more stable to pH 2 than S.boulardii (KT000033)

L 276‐278 – reference to an article where pH effect is studied when there is a bile effect which should
be comment here ‐ review

As per the reviewers valuable suggestion the corrections has been effected in the manuscript

L 314 ‐ anaysis ‐> analysis

As per the reviewers valuable suggestion the corrections has been effected in the manuscript
L317 ‐ put a reference in M&M to control (E. coli OP50)

As per the reviewers valuable suggestion the reference has been incorporated in the manuscript
C. elegans gut colonization assay [20]
L 328 ‐ comments on aggregation when results are not yet presented.

As per the reviewers valuable suggestion the corrections has been effected in the manuscript
L329‐330 ‐ a word is missing? Rephrase the sentence

The outcomes, of the results indicate the  the relationship between attachment ability and cell hydrophobicity. Thus, in vitro studies may not always imitate in vivo situations, which could account for our observation.
L 358 – There is no comment on this paragraph (3.6.2)?

As per the reviewers valuable suggestion the corrections has been effected in the manuscript
L427 ‐ table 2c ‐> suggestion to suppl. Data

The table has been reordered
L430 ‐ paragraph 3‐9 to be revised. Comments on the ability to ferment sugars at different
temperatures are not clear. Lack of explanation on how to calculate growth rates (in fact final Optical
density? So it is not a growth rate).

As per the reviewers valuable suggestion the corrections has been effected in the manuscript
L 464 ‐ review effect of heat treatment: effect on residual population? No effect on growth is
observed but are the inoculation rates (with viable cells) re‐ajusted? To be clarified.

As per the reviewers valuable suggestion the corrections has been effected in the manuscript
L 474 ‐ comment not relevant

As per the reviewers valuable suggestion the corrections has been effected in the manuscript
L 482 ‐ figure 6 ‐ line correspondences are not clear ‐ condition where speed + week is which
condition? (95°C ‐ 90 min? but symbols do not match).

As per the reviewers valuable suggestion the figure error has been edited in the manuscript
L 493 ‐ "heating activity" meaning?

 As per the reviewers valuable suggestion the corrections has been effected in the manuscript

Round 2

Reviewer 1 Report

The review is satisfactory and apart from an improved English check

Reviewer 2 Report

The authors have taken into account the various remarks made and have made the necessary changes.